# Numerical Investigation of a Short Polarization Beam Splitter Based on Dual-Core Photonic Crystal Fiber with As_2_S_3_ Layer

**DOI:** 10.3390/mi11070706

**Published:** 2020-07-21

**Authors:** Nan Chen, Xuedian Zhang, Xinglian Lu, Zheng Zhang, Zhangjian Mu, Min Chang

**Affiliations:** 1Key Laboratory of Optical Technology and Instrument for Medicine, Ministry of Education, University of Shanghai for Science and Technology, Shanghai 200093, China; cn15800968586@163.com (N.C.); 151360021@st.usst.edu.cn (X.L.); zhangzheng2010370@163.com (Z.Z.); 18795901552@163.com (Z.M.); changmin@usst.edu.cn (M.C.); 2Shanghai Key Laboratory of Molecular Imaging, Shanghai University of Medicine and Health Sciences, Shanghai 201318, China

**Keywords:** polarization beam splitter, photonic crystal fiber, extinction ratio, insertion loss, fabrication

## Abstract

A polarization beam splitter is an important component of modern optical system, especially a splitter that combines the structural flexibility of photonic crystal fiber and the optical modulation of functional material. Thus, this paper presents a compact dual-core photonic crystal fiber polarization beam splitter based on thin layer As_2_S_3_. The mature finite element method was utilized to simulate the performance of the proposed splitter. Numerical simulation results indicated that at 1.55 μm, when the fiber device length was 1.0 mm, the x- and y-polarized lights could be split out, the extinction ratio could reach −83.6 dB, of which the bandwidth for extinction ratio better than −20 dB was 280 nm. It also had a low insertion loss of 0.18 dB for the x-polarized light. In addition, it can be completely fabricated using existing processes. The proposed compact polarization beam splitter is a promising candidate that can be used in various optical fields.

## 1. Introduction

A polarization beam splitter [1] (PBS) is an extremely common optical device in optical fiber communication, optical fiber sensing, and optical measuring systems that can split an incident light into two orthogonally polarized lights that constitute a fundamental mode (FM). Usually, a PBS can also be applied in reverse as a beam combiner (BC). Nowadays, there are many waveguides to achieve the polarization beam splitting effect, such as those of the prism, planar waveguide, photonic crystal, metasurface, and optical fiber [2,3,4,5,6]. Among them, fiber-based PBS research is very popular due to its integrability and low cost. However, conventional fibers have an insufficient design flexibility and are glass-based, which is not suitable for excessive processing. Thus the performance of PBSs using conventional fibers as carriers is greatly restricted.

In recent years, photonic crystal fibers (PCFs) have been found to overcome the shortcomings of traditional optical fibers and to greatly broaden the fiber research field. PCFs are widely popular because they have special transmission mechanisms, and their optical properties can be enriched by tailoring the arrangement of internal capillary rods [7,8]. Compared with conventional fibers, PCFs have the unique properties of endless single-mode (ESM) transmission, high birefringence (HB), a large mode area (LMA), and tunable dispersion, so PCFs can be utilized as excellent carriers for in-fiber PBSs. Apart from PBS, common applications include couplers [9] and sensors [10]. Generally, there are two kinds of in-fiber PBSs: One is used to achieve the beam splitting phenomenon by breaking the core symmetry of PCFs for producing the HB effect, and the other is the use of modulation by filling some functional materials into PCFs.

The appearance of PCFs provides the possibility to design new types of in-fiber PBSs. Thus far, scholars have proposed many novel PCF-PBSs. For instance, in 2009, Hameed et al. proposed a PBS based on soft glass PCFs with nematic liquid crystal (LC), and the PBS of a length of 8.227 mm had a cross-talk (CT) of better than 20 dB with bandwidths of 30 nm (TE mode) and 75 nm (TM mode) [11]. In 2011, Zhang et al. created a single-mode single-polarization (SMSP) PCF-PBS by introducing two cores into the SPSM PCF, could can realize the splitting of two communication windows at 1.3 and 1.55 μm [12]. In 2012, Liu et al. proposed a tellurite glass, three-core PCF-PBS with an 8.7983-mm-long splitter; it had an extinction ratio (ER) of better than −20 dB and a bandwidth of 20 nm [13]. In 2013, Sun et al. proposed a dual-core PCF with a metal wire placed into the cladding air hole between the two cores based on the surface plasmon resonance (SPR) effect; the splitter possessed an ER of −20 dB with a bandwidth of 146 nm [14]. In 2014, Chen et al. proposed a novel PBS based on a dual-core PCF with an LC modulation core that had an ultra-broad bandwidth of 250 nm with an ER of better than −20 dB [15]. In 2015, Xu et al. proposed a dual-core PCF with a length of 0.401 mm; here, the ER could reach 110.1 dB at 1.55 μm, and the bandwidths of the ER over 20 and 10 dB could be as wide as 140 and 200 nm, respectively [16]. In 2017, Wang et al. proposed a tunable PBS filled with magnetic fluids (MFs) in air holes; its length was 8.13 mm, and the polarized mode converted at a magnetic intensity of 25 mT with a high ER greater than −100 dB. The tunable PBS could work well at temperatures in the range of 0–55 °C by adjusting the magnetic field strength [17]. In 2018, Younis et al. proposed an asymmetric dual-core PCF, wavelength-selective PBS that can be tuned to split out the x- and y-polarized modes at wavelengths of 1.3 and 1.55 μm [18].

From these PBSs described above, the dual-core PCF is commonly selected as an in-fiber PBS, and the method of filling functional materials such as precious metals, LCs, and MFs into PCFs for generating mode modulation effects is prevalent. In fact, there are many functional materials available for filling PCFs such as rare gases (He, Ne, and Ar), molecular solids (H_2_, CH_4_, N_2_ and O_2_), and some ionic (LiH), covalent (graphite), and metallic (Li and Mg) crystals [19]. The performance of these described PBSs based on the modulation of functional materials is very excellent, and they simultaneously possess a compact size, a higher ER, and a wider bandwidth. Based on these design concepts, we intend to propose a simple-structure, dual-core PCF by coating a ring film with a high refractive index (RI) material—As_2_S_3_—that acts as a PBS. This PCF could be used in the optical fiber communication and sensing field. In our work, the highly applicable finite element method (FEM) [20] was employed to analyze the performance of the compact PCF-PBS by adjusting the structural parameters. Additionally, it is worth mentioning that the proposed PBS can be completely manufactured by modern processes.

## 2. Modeling and Theory

Figure 1a displays the cross-section of the designed PCF-PBS. All air holes were arranged in a hexagonal lattice. The lattice pitch is expressed by *Λ*, and there were only two sizes of air holes, *d*_1_ (*d*_1_ = 1.1 μm) and *d*_2_. Additionally, *d*_2_*/d*_1_ = *α* (*α* is the relative diameter ratio). A thin layer of As_2_S_3_ film was deposited in the central air hole, of which the thickness can be expressed by *t*. The RI of the air was set to 1. The fluoride phosphate (FP) N-FK51A glass was selected as the background material in our design, and its Sellmeier model [21,22] in the investigated wavelength range (1.40–1.75 μm) can be given by:(1)$$n_{N-FK51A}^{2}\left(\lambda \right)=1+\frac{A_{1}\lambda^{2}}{\lambda^{2}-B_{1}}+\frac{A_{2}\lambda^{2}}{\lambda^{2}-B_{2}}+\frac{A_{3}\lambda^{2}}{\lambda^{2}-B_{3}}$$
where *A*_1_ = 0.971247817, *A*_2_ = 0.219014, *A*_3_ = 0.9046517; *B*_1_ = 0.00472302 μm^2^, *B*_2_ = 0.01535756 μm^2^, *B*_3_ = 168.68133 μm^2^, and *λ* is the operating wavelength in the vacuum (in μm).

As_2_S_3_ was selected for the proposed PBS for two reasons. One was its high RI, because a higher RI shows a stronger binding effect on light than the background material. The other reason was its simple deposition process, because the solution-processed As_2_S_3_ glass approach has distinctive virtues over other deposition techniques [23]. Due to the ring structure, special ring modes (RMs) appear on the As_2_S_3_ ring that can be utilized to generate a mode resonance with the dual-core FM for mode classification. The RI of the As_2_S_3_ layer in the central hole can also be determined by a Sellmeier equation [24]:(2)$$n_{As_{2}S_{3}}^{2}\left(\lambda \right)=1+\frac{D_{1}\lambda^{2}}{\lambda^{2}-E_{1}{}^{2}}+\frac{D_{2}\lambda^{2}}{\lambda^{2}-E_{2}{}^{2}}+\frac{D_{3}\lambda^{2}}{\lambda^{2}-E_{3}{}^{2}}$$
where *D*_1_ = 1.898, *D*_2_ = 1.922, *D*_3_ = 0.876, *E*_1_ = 0.022 μm^2^, *E*_2_ = 0.0625 μm^2^, and *E*_3_ = 0.1225 μm^2^.

The effective RI and the mode field distributions for the proposed PBS were solved numerically with the commercial COMSOL5.2 software using the FEM solver. Meanwhile, a perfect matching layer (PML) of 5 μm and a scatter boundary condition (SBC) were set at the outermost layer to improve the accuracy of calculations [25] for the 2D simulation. After doing a discrete transform of the proposed structure using a free triangle mesh, we found 176 vertex elements, 1170 boundary elements, and 18,408 elements in the x–y plane region.

We needed to investigate several main PBS performance parameters. The conventional coupling characteristics of a direction coupler based on a dual-core fiber can be employed to calculate coupling length [12]. The coupling length $L_{c}$ is defined by:(3)$$L_{c}=\frac{\pi }{\left(\beta_{i}^{even}-\beta_{i}^{odd}\right)}=\frac{0.5\lambda }{\left(n_{i}^{even}-n_{i}^{odd}\right)}$$
where i=x, y and $\beta_{i}^{even}$ and $\beta_{i}^{odd}$ represent the propagating constants of the odd and even modes for the x- and y-polarization (x- and y-pol) modes, respectively.

Figure 1b displays schematic diagram of the PBS operation. An incident light carrying the x- and y-pol modes entered the input port with a single mode fiber (SMF). Then different polarized lights were output through different cores with the optical separator. Based on these, we assumed that the light was propagating upon core A. The periodic normalized powers [18] at the output side of cores A and B can be calculated by:(4)$$P_{out}^{A}=P_{in}cos^{2}\left(\frac{\pi }{2}\frac{z}{L_{i}^{c}}\right)$$(5)$$P_{out}^{B}=P_{in}sin^{2}\left(\frac{\pi }{2}\frac{z}{L_{i}^{c}}\right)$$
where $P_{in}$ denotes the input light power assumed to be 1, z denotes the propagation length, and $L_{i}^{c}$ denotes the coupling length for the x- and y-pol modes. According to the coupled mode theory (CMT) [26,27,28,29,30,31], the two polarization states launched into one core of a dual-core PCF can be separated at a specified wavelength when the coupling length of x- and y-pol states satisfies the coupling length ratio (CLR) [32]. The CLR can be defined by:(6)$$CLR=m/n=L_{x}/L_{y}$$
where m/n is equal to even/odd or odd/even. It can be considered that after propagating a length of $mL_{x}$ or $nL_{y}$ in a PBS, two lights with different wavelengths will exit through the A and B out ports.

By implementing an accurate electromagnetic field analysis, the performance of the proposed PBS—comprising the CLR, normalized powers, the ER, and insertion loss (IL)—was investigated.

## 3. Simulation Results and Analysis

When an incident light was transmitted in the proposed PBS, two RMs and four core FMs were investigated, as shown in Figure 2. At the interface between the As_2_S_3_ layer and the substrate material, the x-RM and y-RM were generated, as shown in Figure 2a,b, respectively. Figure 2c–f show the even mode in x-polarization (x-pol EM), the even mode in y-polarization (y-pol EM), the odd mode in x-polarization (x-pol OM), and the odd mode in y-polarization (y-pol OM), respectively, at 1.55 μm. The introduction of a high RI As_2_S_3_ film resulted in a stronger optical coupling between the core FMs, which could change the coupling characteristics of the PCF. It could also be observed that the RM and the FM only coexisted in Figure 2e,f. This phenomenon was similar to the SPR effect between x- and y-pol FMs and the specific second-order surface plasmon polariton (SPP) mode [33,34,35]. The proposed PBS supported two pairs of RI matching between the RM on the As_2_S_3_ film, the core FM could be satisfied at 1.55 μm, and the energy of the core-guided odd mode could couple into the surface of the As_2_S_3_ film. The operation principle of this PBS can be explained on the basis of the modal analysis of the dual-core PCF.

### 3.1. Birefringence and Coupling Length Ratio

Usually, an excellent fiber-based PBS is mainly characterized by the HB effect and a short coupling length. Because of HB, the x- and y-polarization modes in the same core FM cannot easily degenerate, a state that is conducive for beam splitting. Figure 3a shows the birefringence in the proposed PCF with and without the As_2_S_3_ layer. Generally, birefringence [36] can be defined by:(7)$$B=\left|n_{eff}^{x}-n_{eff}^{y}\right|$$
where $n_{eff}^{x}$ and $n_{eff}^{y}$ represent the effective RI for the x- and y-pol modes, respectively; they can be solved based on Maxwell’s equations. From Figure 3a, we can observe that birefringence of the proposed PBS was higher than that of the PBS without the As_2_S_3_ layer. Thus, the proposed PBS had a better splitter effect.

The coupling length can determine the minimum size of a splitter device. According to the report of [14], the choice of the CLR = 2 was conducive to the design of the proposed PCF-PBS because the desired PCF would have a better tolerance to the variation of structural parameters. For PBSs with other kinds of CLRs, when the PCF’s structural parameters change, the device length has larger variations. It is generally difficult to achieve these variations. For simplicity, we supposed the relationship of $L_{x}>L_{y}$, and the dual-core PCF with $L_{x}/L_{y}=2$ at 1.55 μm was able to be achieved for the PBS. We set the PCF length to $L_{x}$, which is the minimum coupling length.

For the CLR, the effect of the characteristic parameters—including the relative diameter ratio *α*, the lattice pitch *Λ*, and the thickness of the As_2_S_3_ layer *t*—was considered. The propagating constants of the even and odd modes of the x- and y-pol lights in the dual-core PCF were obtained from FEM calculation, and then the corresponding coupling lengths were calculated with Equation (3). Finally, the CLR could be deduced. According to Figure 3b–d, when the conditions of *Λ* = 2.2 μm, *d*_1_ = 1.1 μm, *d*_2_ = 2.2 μm, and *t* = 150 nm were simultaneously satisfied, the desired CLR could be achieved. It can be observed from Figure 3b that when the normalized pitch *Λ/**λ* was small, the coupling length increased as the normalized pitch increased. However, after the CLR reached the maximum value of 2 (here *Λ* = 2.2 μm), the CLR decreased with the increase in the normalized pitch. When a variation of ±3% of *Λ* was considered, the corresponding CLR fluctuated around 2, but this fluctuation will be gradually reduced as technology advances and precise structural tailoring can be achieved. Figure 3c displays the effect of the relative diameter ratio on the CLR. When the relative diameter ratio *α* = *d*_2_/*d*_1_ = 2, the coupling length ratio was exactly 2. The perturbation effect of the variation of ±3% of *d*_2_ could be ignored. Finally, as shown in Figure 3d, when the As_2_S_3_ layer thickness was exactly 150 nm, the coupling length ratio was exactly 2. The variation of ±3% of *t* caused the CLR curve to translate, but the maximum value was almost unchanged, thus showing a good fabrication tolerance.

Thus, considering the cases of birefringence and the CLR of 2 in the PCF, these structural parameters of *Λ* = 2.2 μm, *d*_1_ = 1.1 μm, *d*_2_ = 2.2 μm, and *t* = 150 nm were suitable for the design of the desired PBS.

### 3.2. Normalized Power, Extinction Ratio, and Insertion Loss

The normalized power of the x- and y-pol lights in cores A and B for the proposed PCF-PBS regularly changed with the propagation distance. When an incident light travelled a certain distance, the intensity of the polarized light in a certain core was at a maximum value, the intensity of the other polarized light in the corresponding vertical direction was at a minimum value, and two polarized lights were split out. The normalized power also helped us determine the minimum size of the PBS, as shown in Figure 4a. We found that when the length of the PBS was 1.0 mm, only the y-pol and x-pol lights were sustained in cores A and B, respectively, and the separation of the x- and y-pol lights was achieved. We also studied the normalized power of the same structured PBS without the As_2_S_3_ layer. Figure 4b shows that the coupling length achieved by the FEM method was approximately 4.5 mm. Therefore, in terms of compactness, the proposed PBS with the As_2_S_3_ layer was better.

The ER is an important parameter reflecting performance of PBS, and it was employed to describe the degree of polarization separation [37]. $P_{x}^{A}$ and $P_{y}^{A}$ represent the power of the x- and y-pol lights in core A, respectively. For core A, the ER can be defined by:(8)$$ER\left(dB\right)=10log_{10}\left(P_{x}^{A}/P_{y}^{A}\right)$$

The higher the ER is, the better the performance of a PBS is. In practical applications, when the ER reaches −20 dB, the power of one polarized light is 100 times than that of the other polarized light, which is enough to split out the two polarized lights. We regarded the wavelength range corresponding to the extinction ratio of less than or equal to −20 dB as the bandwidth of this PBS. Figure 5 shows the ERs in cores A and B at the 1.55 μm wavelength. As shown in Figure 5a, the minimum ER could reach −83.6 dB, and the bandwidth at 280 nm ranged from 1.45 to 1.73 μm in core A for the proposed PBS. For core B, its minimum ER could reach −49 dB, of which the bandwidth with the extinction ratio less than 20 dB at 160 nm ranged from 1.46 to 1.62 μm. Compared with the proposed PBS, the PBS without the As_2_S_3_ layer had a lower ER and a narrower bandwidth. Core A had a minimum ER of −48.1 dB and a bandwidth of 60 nm. Core B had a minimum ER of −39.3 dB and a bandwidth of 55 nm. These results illustrate that the proposed PCF-PBS had a sufficient ER and a broad bandwidth, and it could stably separate polarized light in the vicinity of 1.55 μm.

Because it is an optical communication device, the IL of the proposed PBS had to be analyzed. Generally, the restriction of light in the core was not perfect, and the IL was one of the potential losses that caused this phenomenon [38]. The IL of the proposed PBS relied on the energy transfer ratio between the two cores and the absorption loss of the As_2_S_3_ layer. Here, the absorption loss of the As_2_S_3_ layer was negligible. Therefore, the IL [39,40] could be defined by:(9)$$IL\left(dB\right)=-10log_{10}\left(\frac{P_{out}}{P_{in}}\right)$$
where $P_{in}$ and $P_{out}$ denote the input power and output power in a core, respectively. For a PBS, the IL should be as small as possible to ensure strong performance. Figure 6 shows various insertion loss values of the x- and y-pol lights for the proposed PCF-PBS with the As_2_S_3_ layer (Figure 6a) and without the As_2_S_3_ layer (Figure 6b) versus wavelength. At the communication window of 1.55 μm, both the x- and y-pol lights had similar IL trends. For the proposed PCF-PBS, the IL was 0.18 dB for the x-pol light and 1.1 dB for the y-pol light. For the PBS without the As_2_S_3_ layer, the IL was 0.55 dB for the x-pol light and 1.35 dB for the y-pol light. Obviously, the proposed PBS had an advantage regarding the IL.

To characterize the performance of the proposed PBS, several key parameters—device size, the ER, bandwidth, and IL—were investigated. Based on the results described above, it can obviously be seen that the performance of the proposed PBS was superior to that of the PBS without the As_2_S_3_ layer.

Additionally, the key performance characteristics of the proposed PCF-PBS and prior PBSs are listed in Table 1 for comparison. From the comparison results, it can be observed that the proposed PBS possessed a compact size, a higher ER, a lower IL, and a broader bandwidth than that of most other PBSs. Though the three PBSs from the literature [41,42,43] had particularly broad bandwidths, they were far less compact than ours. Considering that there are not many PBSs that simultaneous achieve excellent performance for the four parameters, the proposed PBS still has performance advantages and shows great application potential.

## 4. Fabrication Discussion

Additionally, we took the fabrication of the proposed PCF-PBS into account.

The stack-and-draw technique [48,49] can be utilized to fabricate the proposed PCF. For rod stacking, different dimensions of FP glass rods corresponding to the number of cladding holes were stacked into a preform, as shown in Figure 1a. During the drawing process, the preform was drawn into a PCF with the desired dimensions by precisely controlling drawing parameters such as the heat temperature, drawing and perform feed speed, and nitrogen (N_2_) content. In order to ensure the quality of the PCF, the preform surface treatment steps need to be continuously implemented [50].

Next was the integration of the As_2_S_3_ films into the proposed PCF’s central hole. Markos et al. proposed a new fabrication technique for all-fiber nonlinear tunable devices. We adopted such a method to deposit the As_2_S_3_ layer on the inner wall of the central hole. A piece of the proposed PCF was cleaved for filling. Firstly, we dissolved the As_2_S_3_ in an amine solvent. Secondly, to ensure that the solution could pass through the PCF’s central hole, we sealed the air holes in the cladding. except for the central hole, with a UV-curable polymer [51]. Thirdly, we filled the nanogel-like solution into central hole of the PCF via capillary absorption. The filled PCF was placed in an oven at 50 °C for solvent evaporation, and then an amorphous As_2_S_3_ layer was formed in the central hole. Using the annealing process, the optical properties of the PCF with the As_2_S_3_ layer could be adjusted. The thickness of the As_2_S_3_ layer could be monitored by SEM [52,53]. Finally, the end of the PCF blocked by the glue was cleaved off. Using a similar method, the tunable PCF filters and sensing elements based on As_2_S_3_ could also be completely fabricated [54].

Therefore, we believe that the proposed PCF-PBS can be completely manufactured with the support of the described processing techniques.

## 5. Conclusions

In this work, a compact, dual-core PCF-PBS based on a thin As_2_S_3_ film was proposed and numerically demonstrated. When the structural parameters were *d*_1_ = 1.1 μm, *d*_2_ = 2.2 μm, *t* = 150 nm, and *Λ* = 2.2 μm, the relation of *L_x_/L_y_* = 2 could be satisfied at 1.55 μm, which is suitable to design PBSs. Our numerical results showed that the performance of the reported PBS was obviously better than that of a non-As_2_S_3_ PBS with same structure. When the coupling length of the reported PCF-PBS was 1.0 mm, for core A, the ER could reach −83.6 dB, and the bandwidth of the ER better than −20 dB was 280 nm; for core B, the ER could reach −49 dB, and the bandwidth of the ER better than −20 dB was 160 nm. It also had a low IL of 0.18 dB for the x-pol light. Additionally, the proposed PCF-PBS can be completely manufactured by existing processes. This work will help to provide a new idea for designing a dual-core PCF-PBS with a compact size, a high ER, a broad bandwidth, and a low IL. The proposed PBS shows broad application prospects in the communication, sensing, and optical measurement fields.

Future work should conduct the experimental verification of our numerical results. The most direct method of doing this is splicing the PBS to a standard SMF and then launching an incident light into it [55,56]. We plan to build a simple experimental setup to observe the optical characterization of the proposed PBS. Then, we will connect a high power super-continuum source (480–2200 nm), a piece of spliced PCF, an optical separator, a charge-coupled device (CCD) camera, and an optical spectrum analyzer (OSA) in turn. At last, we will compare the collected data of the OSA with the simulation results.

## Figures and Tables

**Figure 1 micromachines-11-00706-f001:**
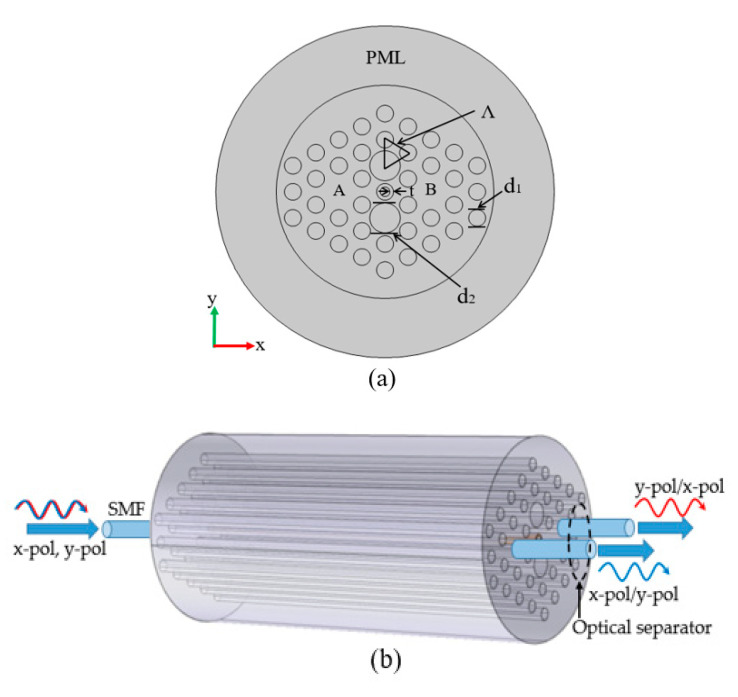
(**a**) Cross-section of the proposed photonic crystal fiber-polarization beam splitter (PCF-PBS). (**b**) Schematic diagram of the PBS operation, where an incident light with the x- and y-pol modes enters the input port, and the x- and y-pol modes can be split out at the output port.

**Figure 2 micromachines-11-00706-f002:**
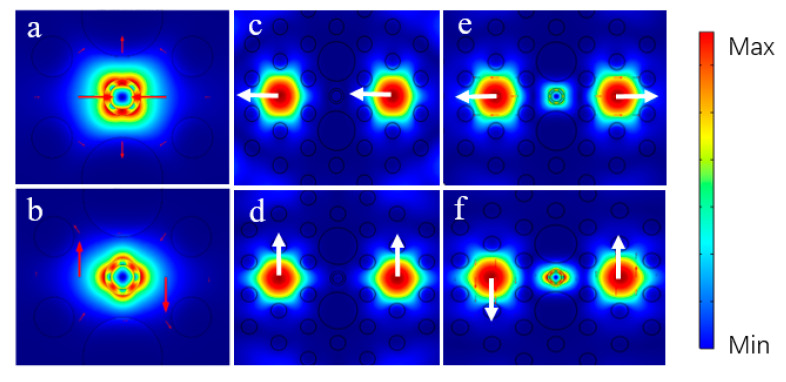
Electric field distributions of the fundamental modes (FMs) for the proposed PCF-PBS at 1.55 μm. (**a**) The x-polarization ring mode (x-RM), (**b**) the y-polarization ring mode (y-RM), (**c**) the x-polarization even mode (x-pol EM), and (**d**) the y-polarization even mode (y-pol EM); the resonance state between the (**e**) x-RM and the x-polarization odd mode (x-pol OM) and the (**f**) y-RM and the y-polarization odd mode (y-pol OM).

**Figure 3 micromachines-11-00706-f003:**
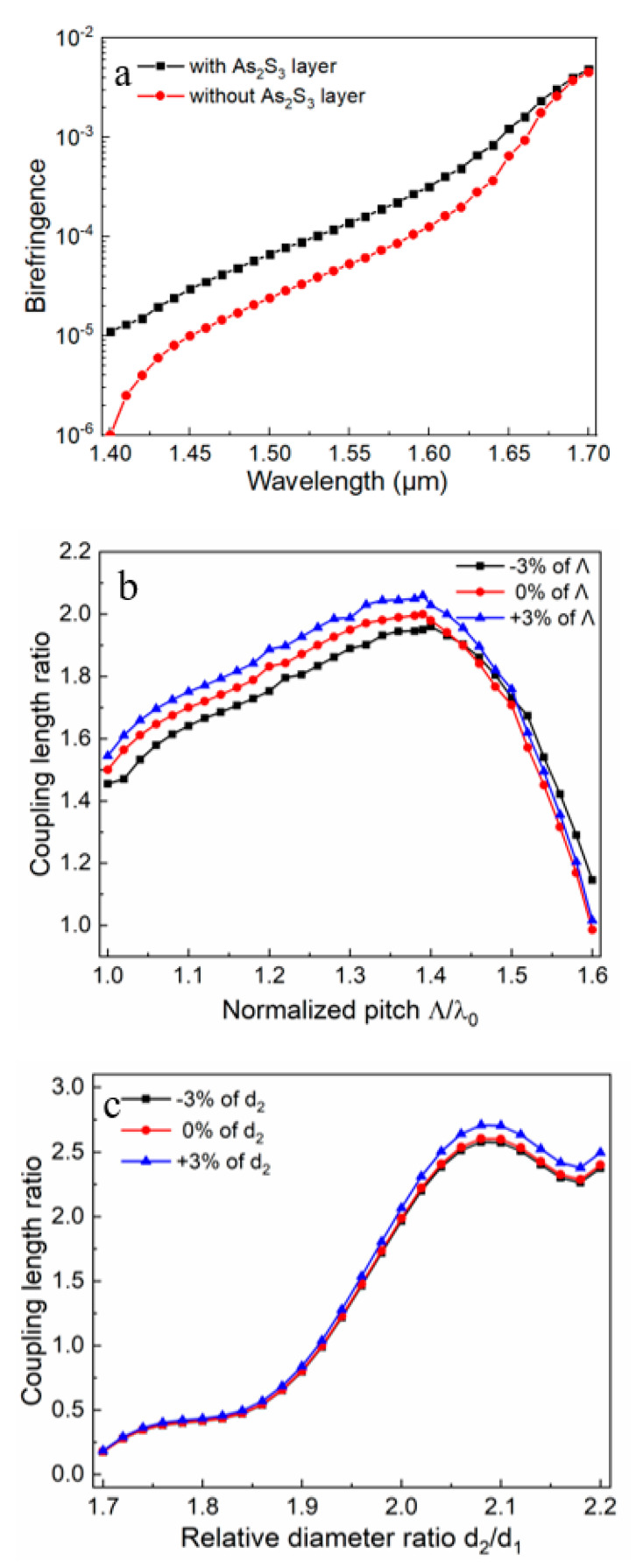
(**a**) Birefringence in the proposed PCF-PBS with and without the As_2_S_3_ layer, as well as the effect of different PCF characteristic parameters on the coupling length ratio (CLR) for the proposed PBS. (**b**) The CLR versus normalized pitch when *d*_1_ = 1.1 μm, *d*_2_ = 2.2 μm, and *t* = 150 nm; (**c**) the CLR versus the relative diameter ratio when *Λ* = 2.2 μm, *d*_1_ = 1.1 μm, and *t* = 150 nm; and (**d**) the CLR versus the As_2_S_3_ layer thickness when *Λ* = 2.2 μm, *d*_1_ = 1.1 μm, and *d*_2_ = 2.2 μm.

**Figure 4 micromachines-11-00706-f004:**
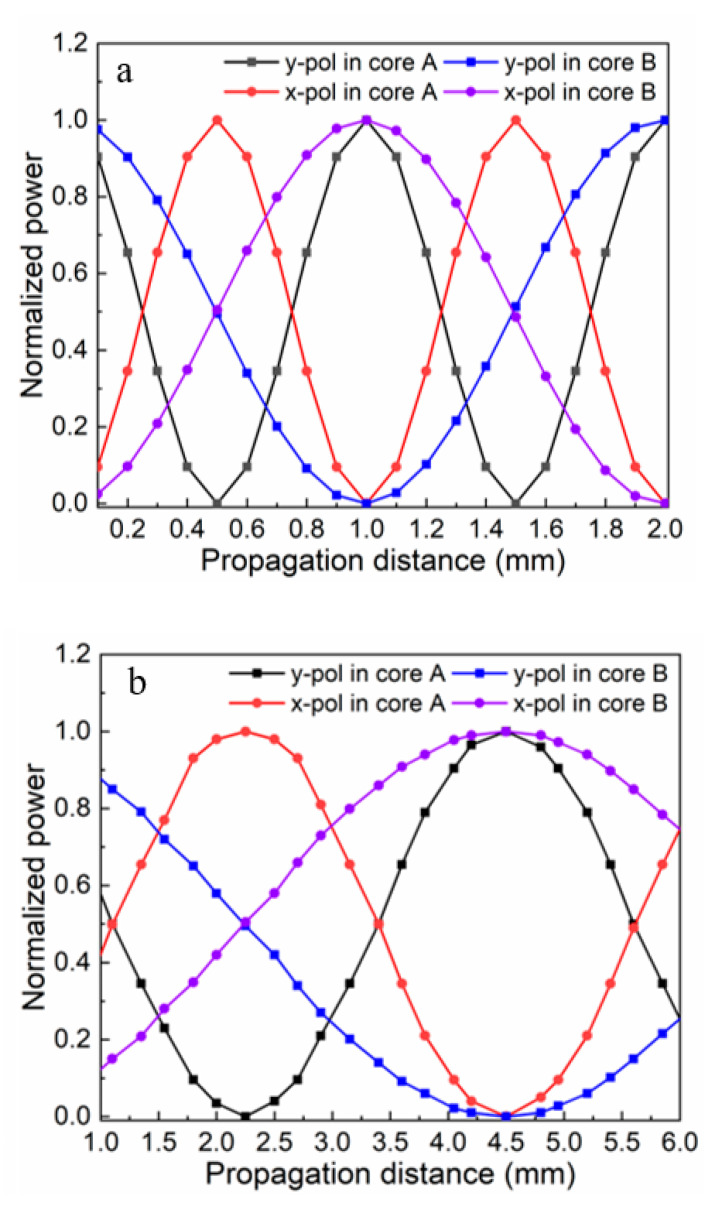
Various energy transfer values in cores A and B for the proposed PCF-PBS (**a**) with the As_2_S_3_ layer and (**b**) without the As_2_S_3_ layer versus the propagation distance when *Λ* = 2.2 μm, *d*_1_ = 1.1 μm, *d*_2_ = 2.2 μm, and *t* = 150 nm.

**Figure 5 micromachines-11-00706-f005:**
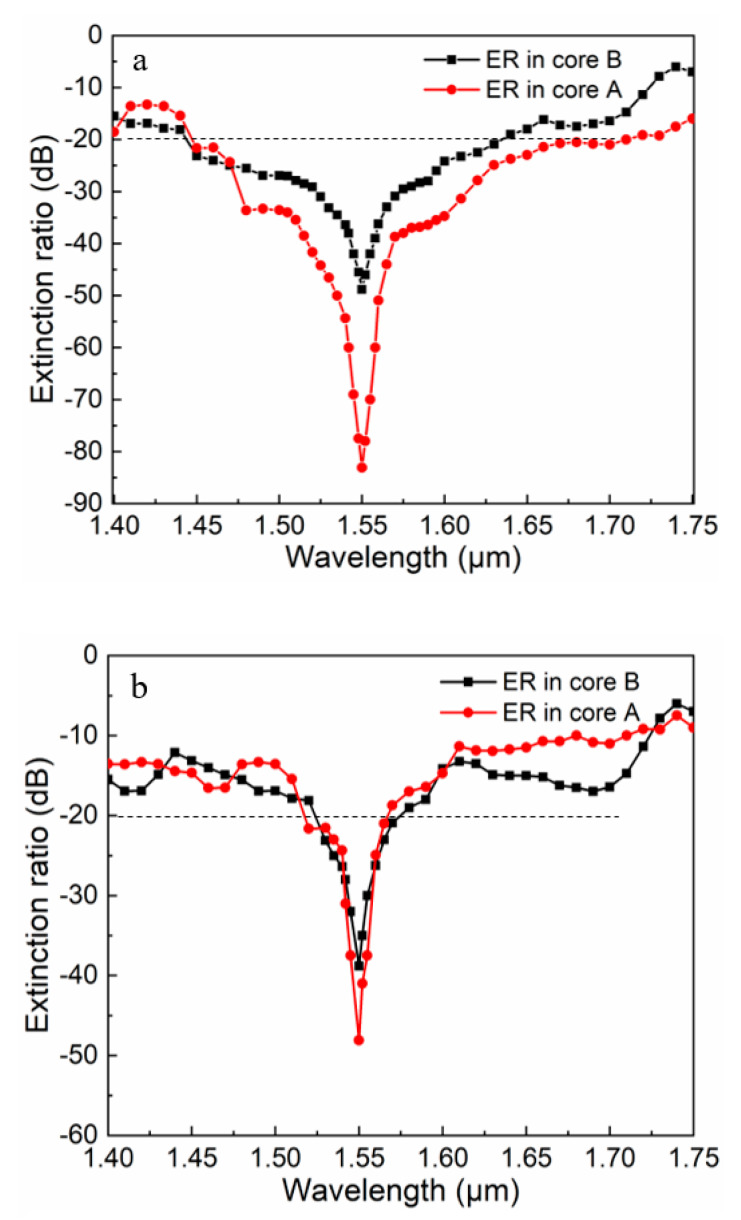
Various extinction ratios (ERs) of the x- and y-pol lights for the proposed PCF-PBS (**a**) with the As_2_S_3_ layer and (**b**) without the As_2_S_3_ layer versus wavelength when *Λ* = 2.2 μm, *d*_1_ = 1.1 μm, *d*_2_ = 2.2 μm, and *t* = 150 nm.

**Figure 6 micromachines-11-00706-f006:**
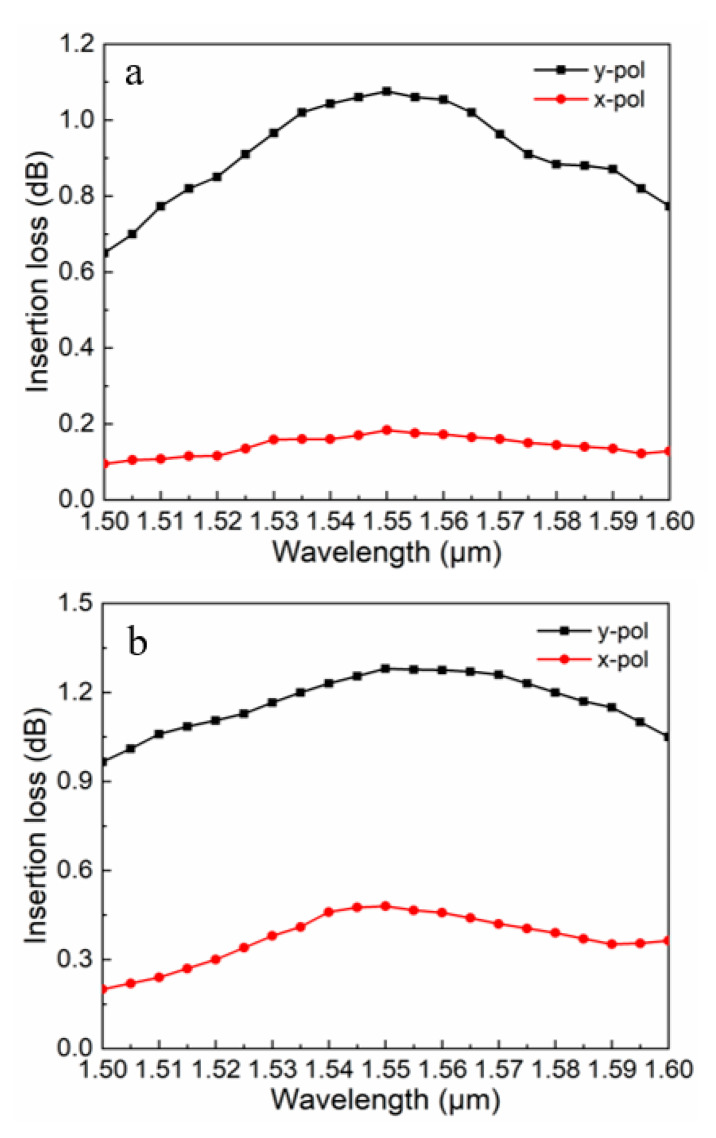
Various insertion loss (IL) values of the x- and y-pol lights for the proposed PCF-PBS (**a**) with the As_2_S_3_ layer and (**b**) without the As_2_S_3_ layer versus wavelength when *Λ* = 2.2 μm, *d*_1_ = 1.1 μm, *d*_2_ = 2.2 μm, and *t* = 150 nm.

**Table 1 micromachines-11-00706-t001:** Comparison results between the proposed PBS and prior PBSs.

Ref	Device Length (mm)	ER (dB)	Bandwidth (nm)	IL (dB)
[13]	8.7983	−164.2681	20	N/A
[14]	~6	−40	146	N/A
[15]	0.09	−80.7	250	N/A
[16]	0.401	−110	140	N/A
[17]	8.13	>−100	~20	N/A
[18]	5.678	−30	3	N/A
[31]	2	−52.5	100	N/A
[33]	0.3	23	~20	N/A
[41]	84.7	−30	300	N/A
[42]	52.8	~50	320	N/A
[43]	72.5	~50	400	N/A
[44]	1.9	~−35	37	N/A
[45]	1.9207	−65	226	0.42737
[46]	0.103	−72	177	0.00013
[47]	0.078	87	40 (>15 dB)	N/A
The paper	1.0	−83.6	280	0.18

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
