# Peer review of "Numerical Investigation of a Short Polarization Beam Splitter Based on Dual-Core Photonic Crystal Fiber with As2S3 Layer"

_micromachines, 2020, doi:10.3390/mi11070706_

Round 1

Reviewer 1 Report

I have revised the manuscript entitled "Numerical Investigation of a Short Polarization Beam Splitter based on Dual-Core Photonic Crystal Fiber with As2S3 Layer" and here are my comments.
The presented topic devoted to polarization beam splitter based on PCF is very interesting and is presented clearly.
The overall quality of the paper is very high. The text is well written in English (without grammar mistakes).
I recommend publication of this paper in the "Micromachines" journal after addressing one issue.

1. Why did the authors use As2S3 material to infiltrate the central hole of the proposed PCF?
What are the advantages of using this material? Moreover, did the authors considered using other materials instead of As2S3?
In my opinion, the authors should include some explanation of using As2S3 into the text. That would enrich the quality of the paper.

Author Response

I have revised the manuscript entitled "Numerical Investigation of a Short Polarization Beam Splitter based on Dual-Core Photonic Crystal Fiber with As2S3 Layer" and here are my comments.
The presented topic devoted to polarization beam splitter based on PCF is very interesting and is presented clearly. The overall quality of the paper is very high. The text is well written in English (without grammar mistakes). I recommend publication of this paper in the "Micromachines" journal after addressing one issue.

Comment 1: Why did the authors use As2S3 material to infiltrate the central hole of the proposed PCF? What are the advantages of using this material? Moreover, did the authors considered using other materials instead of As2S3? In my opinion, the authors should include some explanation of using As2S3 into the text. That would enrich the quality of the paper.

Response: Thank you for your approval. Firstly, the device has the highest coupling efficiency when using As2S3 material to infiltrate the central hole of the proposed PCF. Secondly, the solution-processed approach of chalcogenide glass has discrete advantages over other deposition techniques such as simple preparation procedure, deposition of thin or thick films on complex and non-planar structures, integration among different optical devices, etc. In contrast, the process of depositing metal in the hole is more difficult. Thirdly, we think that as long as it has similar properties and similar resonance effects, As2S3 can be replaced by other materials for designing PBS. We have added relevant statements that marked in red on L89-L92 in Introduction. Thank you for your suggestions again.

Author Response

Title: Numerical Investigation of a Short Polarization Beam Splitter based on Dual-Core Photonic

Crystal Fiber with As2S3 Layer. Authors: Nan Chen et al. Manuscript ID: micromachines-854515.

In this work, the authors numerically demonstrate a dual-core photonic crystal fiber infiltrated with chalcogenide arsenic trisulfide (As2S3) glass film and they show how the final device acts as polarization beam splitter. The authors used the very well-known finite element method to calculate the guiding properties of this fiber. A polarization extinction ration of -83.6 dB can be achieved at 1.55 μm wavelength over 1 mm length.

In general, the numerical part of the manuscript seems consistent and the figures are well-presented.

It is true that there is a huge amount of numerical works on dual-core fibers for sensing and other applications including polarization beam splitting but also “hybrid fibers” by combining the existing PCF technology with advanced functional materials. I could suggest the publication of the current manuscript to Micromachines journal but there are still unclear claims and significant part of missing literature that the authors should convincingly address prior any acceptance. In particular:

Comment 1- In the introduction the authors state: “Compared with conventional fibers, PCFs have the unique properties of endless single-mode (ESM) transmission, high birefringence (HB), large mode area (LMA), tunable dispersion, etc. So PCF can be utilized as an excellent carrier for an in-fiber PBS”. I agree, but dual-core PCF are serving many other purposes apart PBS such as for example the highly sensitive fiber biosensors using a dual-core PCF [A]. The authors should carefully refer and cite the prior literature.

Response: First of all, thank you for your approval. We have added some statements about other purposes apart PBS that have been marked in red on L42-L43 in Introduction. Thank you for your suggestions.

Comment 2- Also in the introduction, the authors state: “…dual-core PCF is commonly selected as an in-fiber PBS and the method of filling functional materials such as precious metals, LCs, MFs and others into PCFs for generating mode modulation effects is prevalent…”. There is also a large amount of literature in the field of material-filled PCFs and I suggest to the authors to refer/cite a relatively recent extensive review article in this topic [B].

Response: Thank you for your guidance. We have added some statements about functional materials which have been marked in red on L68-L71 in Introduction.

Comment 3- Why the authors decided to use the stoichiometric As2S3 and not the selenide-based glass As2Se3? The refractive index of the latter is higher than the former and thus a stronger coupling is expected. The authors should elaborate on their choice.

Response: Thank you for your comments. Our original plan was to design an in-fiber PBS different from the traditional PCF-PBS and metal-based PCF-PBS. We suppose that As2Se3 would also work according to the theory in our paper. The As2Se3-based PBS is a good scheme. We intend to explore the feasibility of this scheme in future.

Comment 4- The authors state that “The RI of As2S3 layer in central hole can also be determined by a Sellmeier equation [20].” Looking ref 20. is not clear to me whether the material dispersion used in this work as well is based on measured data. I suggest to the authors to also consider Ref. [C] and the dispersion profile of As2S3 used in their study just to be sure that the dispersion profile is correct. This parameter will significantly affect the final results.

Response: Thank you for your comments. We are sorry that the Sellmeier model is not clear. The model refers to the report [24] (Saghaei H.; Heidari V.; Ebnali-Heidari M.; Yazdani, M. R. A systematic study of linear and nonlinear properties of photonic crystal fibers. Optik 2016, 127, 11938-11947.). We have made corrections which have been marked in red on L349-L350 in Reference.

Comment 5- Did the authors consider in their calculations the material loss of As2S3 in the telecom region? They state “The IL of the proposed PBS relies on the energy transfer ratio between the two cores and the absorption loss of As2S3 layer” However, it is not clear whether they included or not the material loss of the chalcogenide film? If so, could they provide a value?

Response: Thank you for your comments. In our works, the IL is calculated by the energy transfer ratio and the absorption loss is negligible. We have corrected the relevant statements that marked in red on L233-L234 in Section 3. We didn’t consider the material loss of As2S3. In fact, the material loss mainly includes absorption loss and scattering loss. Your suggestion is very valuable, so we will consider these factors in the next work. Thank you for your guidance again.

Comment 6- In the fabrication discussion, the authors discuss the experimental feasibility of their device. The chalcogenide glass film integration is relying upon a previous approach using a solution-process. For completeness purposes, I suggest to the authors to also consider which the same approach has been developed for tunable sensing PCF devices.

Response: Thank you for your suggestions. We have added some statements that have been marked in red on L277-L278 in Section 4.

Comment 7- It is not clear how the selectively filling process will be achieved. The authors state “…Secondly, seal the air holes in the cladding with glue except the center hole.” I agree but they should be more specific how this will be experimentally achieved and provide perhaps a reference.

In general, the current manuscript can be accepted for publication if the authors address all the aforementioned major comments. I would be happy to see the revised manuscript.

[A] Opt. Express 19, 7790-7798 (2011).

[B] Rev. Mod. Phys. 89(4), 045003 (2017).

[C] J. Lightwave Technol. 33(13), 2892–2898 (2015).

[D] Sci. Rep. 6(1), 31711 (2016).

Response: Thank you for your comments. This reference about sealing the air holes in the cladding has been added and cited on L270-L272 in Section 4. Thank you for your guidance again.